

This is a non-peer-reviewed preprint. This manuscript is submitted for publication in Solid Earth Journal. Subsequent versions of this manuscript may differ slightly in content. Once accepted, the published version will be made available through the ‘peer-reviewed publication doi’ link on the right-hand side of this webpage. Please feel free to contact any of the authors;

we welcome feedback



**Impact of stress regime change on the permeability of a naturally fractured carbonate buildup (Latemar, The Dolomites, Northern Italy)**

Onyedika Anthony Igbokwe<sup>1,2,3\*</sup>, Jithender J. Timothy<sup>4</sup>, Ashwani Kumar<sup>5</sup>, Xiao Yan<sup>6,7</sup>, Mathias Mueller<sup>3</sup>, Alessandro Verdecchia<sup>3</sup>, Günther Meschke<sup>7</sup>, Adrian Immenhauser<sup>3,8</sup>

Affiliation:

<sup>1</sup>School of Earth and Environmental Sciences, University of St Andrews, KY16 9AL, UK

<sup>2</sup>~~Department of Physics, Geology and Geophysics, Alex Ekwueme Federal University Ndifur-Alike, Ikwo, P.M.B. 1010, Abakaliki, Ebonyi State, Nigeria~~

<sup>3</sup>~~Ruhr-University Bochum, Institute of Geology, Mineralogy and Geophysics, Universitätsstraße 150, 44801 Bochum, Germany~~

<sup>32</sup>~~Department of Physics, Geology and Geophysics, Alex Ekwueme Federal University Ndifur-Alike, Ikwo, P.M.B. 1010, Abakaliki, Ebonyi State, Nigeria~~

<sup>4</sup>~~Centre for Building Materials, Technical University of Munich, Franz-Langinger-Straße 10, 81245, Munich, Germany~~

<sup>5</sup>~~Advance Manufacturing Lab, ETH Zürich, Switzerland~~

<sup>6</sup>~~Department of Geotechnical Engineering College of Civil Engineering, Tongji University, Shanghai 200092, China~~

<sup>7</sup>~~Ruhr University Bochum, Institute for Structural Mechanics, Universitätsstraße 150, 44801 Bochum, Germany~~

<sup>8</sup>~~Fraunhofer IEG (Institution for Energy Infrastructures and Geothermal Systems), Am Hochschulkampus 1, 44801 Bochum, Germany~~

\* Corresponding author Email: [oai1@st-andrews.ac.uk](mailto:oai1@st-andrews.ac.uk)

**Abstract**

Changing stress regimes control fracture network geometry and influence porosity and permeability in carbonate reservoirs. Using outcrop data analysis and a displacement-based linear elastic finite element method, we investigate the impact of stress-regime change on fracture network permeability. The model is based on fracture networks, specifically fracture 65 sub-structures. The Latemar, predominantly affected by subsidence deformation and Alpine compression, is taken as an outcrop analogue for an isolated (Mesozoic) carbonate buildup with fracture-dominated permeability. We apply a novel strategy involving two compressive boundary loading conditions constrained by the study area's NW-SE and N-S stress directions. Stress-dependent heterogeneous apertures and effective permeability were computed in 2-D 70 domain by (i) using the local stress state within the fracture sub-structure and (ii) running a single-phase flow analysis considering the fracture apertures in each fracture sub-structure. Our results show that the impact of the modelled far-field stresses at (i) subsidence deformation from the NW-SE and (ii) Alpine deformation from N-S increased the overall fracture aperture and permeability. In each case, increasing permeability is associated with open fractures 75 parallel to the orientation of the loading stages and with fracture densities. The anisotropy of permeability is increased by the density and connectedness of the fracture network and affected by shear dilation. The two far-field stresses simultaneously acting within the selected fracture sub-structure at a different magnitude and orientation do not necessarily cancel out each other in the mechanical deformation modelling. These stresses affect the overall aperture and 80 permeability distributions and the flow patterns. These effects—potentially ignored in simpler stress-dependent permeability—can result in significant inaccuracies in permeability estimation.

## 1. Introduction

Naturally fractured reservoirs accommodate a significant share of the world's hydrocarbon 85 resources, especially in carbonates that contain 50 to 60% of oil and gas reserves worldwide (Garland et al., 2012). These reservoirs also play an essential role in transitioning to a low-carbon energy future, especially in producing low- and high-enthalpy geothermal heat or sequestering CO<sub>2</sub> (McNamara et al., 2015). However, these potentials have not been fully

explored partly because of the challenge of predicting and quantifying the contribution of fluid flow through fractures in naturally occurring complex fracture networks (Berkowitz, 2002; Narr et al., 2006). The challenge is primarily due to the sub-seismic heterogeneous characteristics of fractures, partially studied from core data or image logs (Laubach, 2003), and the lack of an understanding of the structural arrangement of the fracture networks and their 3-D distribution and geometrical attributes. Given that core data or image log (capturing small-scale or local information around a well) lacks information on the spatial arrangement (Bourbiaux et al. 2005), outcrop analogues are often used for explicit descriptions of fracture distributions, including length, orientation, spacing and aperture (Agosta et al., 2010; Bonnet et al., 2001; Hooker et al., 2013; Igbokwe et al., 2018, 2020; Wilson et al., 2011). Integrated and detailed outcrop studies provide constraints for understanding most fracture geometrical parameters because analogues have adequate resolution over a varied length scale of  $10^{-1}$  to  $10^4$  m, even in three dimensions (Rotevatn et al., 2009; Igbokwe et al., 2020). Also, outcrop analogues can be extrapolated to reservoir scale fracture models based on geomechanical relations as long as the data are accurately captured and corrected for sampling artefacts (Bisdom et al., 2016a; Narr et al., 2006).

In many natural reservoirs, the fractures act as the principal pathways for fluid or gas flow, particularly in low matrix permeability rocks. Fracture aperture is one of the main factors controlling fracture flow, as the aperture delineates fracture porosity and permeability (Bisdom et al., 2016b; Hooker et al., 2013). Minor variations in aperture have enormous implications on rock flow and transport properties (de Dreuzy et al., 2002; Matthäi, 2003; Matthäi & Belayneh, 2004), and obtaining a realistic aperture prediction from outcropping geometries is challenging. At depth, *in-situ* stresses partly control the aperture, which influences permeability by either increasing or decreasing their order of magnitude (Lei et al., 2015, 2017; Zoback, 2007), but pressure relief during exhumation and meteoric diagenesis (specifically rainwater) may leach and even dissolve cement-occluded fractures and change the aperture. Thus, the outcropping aperture cannot be considered representative, except when these apertures are generated from veins that have not been reactivated during exhumation (e.g., Hooker et al., 2013, 2014). That said, preserved veins covering a large outcrop scale equivalent to a reservoir scale are rarely found in nature.

As an alternative, fracture aperture is modelled as a function of stress (shear stress-induced dilations), using hydromechanical coupling based on linear elastic fracture mechanics (Bisdom et al., 2017; Lei et al., 2015; Min et al., 2004). These models require the local stress state, typically derived from Finite Element (FE) models with explicit fracture representations

(Bisdom et al., 2016a; Lei et al., 2017). The local state of stress can change the hydraulic properties and fluid pressure and vice versa in a rock domain. For stress-induced changes in 125 hydraulic properties, permeability can be several orders of magnitude greater and irreversible under perturbations resulting from various natural activities. These natural activities can cause stress redistribution, such as in geothermal energy and oil/gas ~~reservoir operations~~<sup>production</sup>, where injections and extractions of fluids demand ~~meaningful~~<sup>significant</sup> change in effective 130 stresses ~~in the subsurface~~<sup>underground</sup> (e.g., Chen & Teufel, 2001; Min et al., 2004). Thus, investigating the impact of stress change on permeability becomes fundamental in understanding the reservoirs' overall aperture distribution and flow pattern.

The effect of stress on the permeability of fractured rocks has been widely investigated using synthetic fracture networks and 2-D fracture network models. For example, differential stress 135 considerably impacted the magnitude and direction of rock mass permeability when the stress effects on the 2-D permeability tensor of three sampled natural fracture networks were analysed (Zhang & Sanderson, 1996). Min et al. (2004) observed significant stress-induced flow enhancement along with connected shear fractures in the study of stress-dependency of rock mass permeability with the effects of nonlinear joint normal deformation and shear dilation. While Bisdom et al. (2016a) investigated the influence of *in-situ* stress on the permeability of 140 an outcrop-based fracture system, Lei et al. (2014) analyzed the stress effect on the validity of synthetic fracture networks for representing a naturally fractured rock in terms of geomechanical and hydraulic properties.

These previous studies mainly considered the tectonic stress perturbation based on a single stress regime that scrutinises the magnitude and orientation of principal stresses on a rock body 145 differently. However, past studies did not necessarily consider the different tectonic episodes that build up different stress regimes and which may change in geological time. In nature, the stress in a rock body can exist in two forms: (i) the far-field stress impacting the rock from outside the body and (ii) the local state of stress domiciled inside the rock body. The far-field stress is of tectonic origin or related to lateral pressure changes affected by the lateral thickness 150 and density variations (Pascal & Cloetingh, 2009). At the same time, the local state of stress is characterised by rotations and changes in principal stress orientations, magnitude and other forces acting on the rock body. The local state of stress changes is linked to the impact of far-field stresses and geometry, distribution and density of fracture network characteristics. Therefore, investigating stress perturbation and its effects on the permeability of rock bodies 155 needs a comprehensive approach that will capture the impact of the magnitude and orientation

of the far-field stresses and the associated changes in the local state of stress in different tectonic episodes.

The difficulty is primarily representing (i) the stress-regime changes over time, (ii) the complex fracture system geometry, having various orientations and lengths, and (iii) the complex mechanical deformation mechanism, influencing the interactions between individual fractures in an FE mechanical model. Until now, a focused study dealing with the superposition of different magnitudes and directions of the far-field stresses, representing the major geological tectonic episodes in a given area, ~~has not been well represented~~ in several FE mechanical models has been hampered. In complex fractured rock, the interactions between the rock matrix and fractures present the primary challenge for FE in solving the hydro-mechanical coupling problem. The fractured rock are generally assumed as a continuum medium with dual porosity and permeability, while the real geometry and hydro properties cannot be described accurately (Hoteit & Firooazbadi, 2006; Ren et al. 2017). This is in addition to the complicated mathematical calculation. As such, the overall impact of stress-regime change on permeability remains poorly understood.

In this study, we investigate the influence of the superposition of two orthogonal far-field stresses derived from significant tectonic episodes on the permeability of carbonate rocks with different fracture distributions. The natural fracture geometry was utilised with multiple fracture sets and intersections from an outcrop analogue at the Latemar carbonate buildup. The two major tectonic events at Latemar are associated with subsidence-related deformation in Late Triassic and Early Jurassic times, shortly after the fractures were formed, and later Alpine compression during the Neogene. Although new fractures may have formed during these tectonic episodes, we assumed that new fractures did not form and/or grow in the model used in this study. Instead, already-developed fractures were either opened and/or closed during these tectonic events. This enables us to study stress regime change effects on permeability, focusing on those issues related to the complexity of multiple fractures. The development of the Latemar fractures and their driving factors, including their geometries, connectivities, crosscutting relationships etc., is beyond this paper's scope and has been discussed in detail in Igobokwe et al. (2022). We aim to draw general conclusions about the impact of stress regime change on permeability in complex fractured systems, which representing -an analogue of ~~the~~ subsurface reservoirs.

The main objectives of this paper are to (i) analyse the selected fracture network characteristics of the study area, (ii) evaluate the impact of stress regime change, considering the overall tectonic episodes, (iii) compute the stress-induced fracture aperture in a FE model,

and to (iv) evaluate the changes in the effective permeability and permeability anisotropy. The implication of the central assumptions and the impact of the changing stress orientation and magnitude on effective permeability are discussed.

## 2. Geological background and study area

The Latemar carbonate buildup is one of the pre-volcanic Middle Triassic isolated carbonate platforms (Goldhammer & Harris, 1989; Preto et al., 2011) located in the southwestern part of the Dolomite Mountain belt (Northern Italy; Fig. 1) and neighboured by the Catinaccio and the Agnello platforms in the north and south, respectively (Fig. 1a). The outcrops of the Latemar are built predominantly by the Sciliar (Schlern) Formation (Fig. 1b), underlying the Contrin Formation, which is a regionally important carbonate bank in the Dolomite Mountain belt 200 (Gaetani et al., 1981; Jacquemyn et al., 2014). Between the Late Anisian and Late Ladinian (Middle Triassic), the Latemar buildup formed on topographic highs. Their buildups were separated by basinal areas where siliceous basinal carbonates were deposited (Fig. 1; Bosellini, 1984).

The Latemar has a maximum altitude of 2850 m, with high peaks exposing the platform margin, the slope and the interior (Fig. 1c, d). While the preserved portions of the platform margin consist of reefal boundstones, microbial crust, and marine cement-facies, the slope is characterised by massive breccia flows, including coarse and matrix-poor materials derived from the platform margin or platform interior (Egenhoff et al., 1999; Emmerich et al., 2005; Goldhammer & Harris, 1989; Harris, 1994; Marangon et al., 2011). The platform interior is 210 arranged in dm's-to m's-scale shallowing upward cycles (Christ et al., 2012; Goldhammer et al., 1990), consisting of up to 750 m successions of subtidal and peritidal carbonate lagoonal deposits. The Latemar buildup was partly dolomitised due to fluid mobilisation triggered by the Predazzo Volcanic-Plutonic Complex in the Middle Triassic. A recent review on the dolomitisation and diagenesis of the Latemar buildup is given by Mueller et al. (2021).

The considered outcrop in the presented study area is a large fractured pavement in the platform interior (Fig. 1c, d). The studied pavement (*ca.*  $7.5 \times 10^2$  m<sup>2</sup>) consists of limestone and dolostone rock bodies, showing a gentle dip of fewer than 5° towards the north. Most structural features observed on the fractured pavements are fractures, veins and stylolites, some of which are weathered.

### 220 2.1. Tectonic setting

The Latemar forms part of the Southern Alps, part of the Mesozoic Adriatic plate, predominantly thrusting southward during the Alpine collision (e.g., Boro et al., 2013; Doglioni, 1988). Deposition in the Latemar began on a structural high (horst) generated by extensional tectonics, breaking up the widespread regional carbonate bank (Contrin Fm.).

Subsequently, subsidence deformation and extensional synsedimentary tectonics controlled the geometry of the buildup, leading to faulting. Several fractures and/or faults crosscut the Latemar buildup, formed in conjugate pairs, and are related to deformation induced by Middle Triassic subsidence. Preto et al. (2011) documented the ENE-WSW and WNW-ESE faults linked to the Triassic trans-tensional regime as the oldest fault direction in the Latemar. The

outcome is a buildup with a horseshoe shape and intraplatform basins (Fig. 1a, c; Preto et al., 2011). Dextral strike-slip reactivation, observed along magmatic dikes, reflects Neogene Alpine compressional tectonics in the Latemar.

*Figure 1. (a) Overview of the Latemar and the neighbouring Ladinian carbonate buildups (or platforms) and Upper Ladinian intrusions of the Dolomites (modified after Jacquemyn et al., 2014, 2015). (b) Simplified stratigraphic chart of the Dolomite Mountains, modified after Jacquemyn et al., 2014. (c) Drone image showing the general overview of the horse-shoe shape of the Latemar buildups. The geometry and topology of the outcrops pointed with arrows have been presented in Igboekwe et al. (2022). (d & e) Drone images acquired from the outcrop pavement at platform interior. The red arrows show images of a person (for scale) with a wide feet length of 1m and a hight of approximately 2m*

Before the Neogene compressional tectonics, a regional magmatic-tectonic event in the Late Ladinian to Early Carnian triggered intense magmatic activity. These generated the intrusion of the Predazzo Volcanic-Plutonic Complex and the Mt. Monzoni, which are associated with temporarily halting carbonate deposition at Latemar (Bellieni et al., 2010; Bosellini, 1984; Bosellini et al., 2003).

In the Latemar buildup, two principal far-field stresses did affect the carbonate deposits, representing the two main phases of deformation, namely NW-SSE (Middle Triassic extensional (subsidence) tectonics) and the N-S (Alpine compressional tectonics) (Boro et al., 2013; Hardebol et al., 2015; Jacquemyn et al., 2015; Preto et al., 2011).

### 3. Data and Methods

The methods applied in this study include a three phase workflow: (i) fieldwork, including drone image acquisition and outcrop interpretation and digitisation (Fig. 2a), (ii) meshing and geomechanical finite element aperture modelling and calculations (Fig. 2b) and (iii) fluid-flow modelling and effective permeability calculations for uncoupled hydromechanical conditions.

**260**

**Figure 2. Workflow for obtaining flow-base effective permeability using acquired drone images from outcropping fracture networks. a) (Phase 1.1) Drone imagery, photogrammetry and (Phase 1.2) fracture interpretation and digitization (the illustration shown is a Fracture Sub-Sample (FSS) with ~ 2 x 2 m dimension). b) (Phase 2.1) Simplifying and converting the interpreted fracture network geometry to element geometry and, then, meshing, (Phase 2.2) local stress modelling, (Phase 2.3) geomechanical finite-element fracture-aperture modelling and calculations. c) (Phase 3.1) Fluid-flow modelling, considering "uncoupled" conditions and (Phase 3.2) effective permeability calculations.**

### 3.1. Field data and fracture network geometry

**265** Field investigations of Latemar outcrops shows widespread brittle deformation features with low-strain barren fractures and veins predominantly with sub-vertical dip as the dominant structures. These features, including bedding-parallel and bedding-perpendicular (tectonic)

stylolites, were photographed, mapped, and structurally characterized within the exposed outcrop stations, and in general, are displayed as mode I, and conjugate hybrid fractures<sup>270</sup> and stylolites.

By considering the prominent sub-vertical outcrops exposed at the base of the Latemar (in the Valsorda Valley; Fig. 3) and sub-horizontal (pavement) outcrops at the flat-topped Latemar buildup (Fig. 4), the arrangements orientations and the stress fields during the development of the fractures are documented. The stress state, constraining the deformation, is reconstructed<sup>275</sup> from the fault-slip data, including the strike and dip of the fault plane, the orientation of the slip line and the shear sense on the fault plane. This reconstruction is based on the assumption that the remote stress tensor is spatially uniform for the rock mass containing the faults and temporally constant over the history of faulting in the given region, and the slip on each fault surface has the same direction and sense as the maximum shear stress resolved on that surface<sup>280</sup> (Bott, 1959; Kaven et al., 2011). The inversion of these fault-slip data gives the principal stress axes parameters,  $\sigma_1$  (maximum compression),  $\sigma_2$  (intermediate compression) and  $\sigma_3$  (minimum compression; Delvaux & Sperner, 2003).

In the Valsorda Valley (Fig. 3), carbonate outcrops are affected by minor reverse conjugate faults dipping at low angle ( $< 30^\circ$ ) to bedding. These conjugate reverse faults strike between ca  $238^\circ$  WSW-ENE and  $250^\circ$  SW-NE, accommodating low-angle SSE - and ENE dipping fractures with a horizontal intersection. Their kinematic indicators and movements point to dip-slip motion. Besides the conjugate reverse fault, bedding-parallel and bedding-perpendicular stylolites were observed. The bedding-perpendicular stylolites are oriented primarily on ca  $230^\circ$  to  $250^\circ$  NE-SW to ENE-WSW and strike perpendicular to the maximum sub-horizontal stress.<sup>285</sup> These stylolites coupled with the orientation of low-angle conjugate reverse faults, determined an approximate NNW-SSE trending sub-horizontal  $\sigma_1$  stress (Figs. 3). Overall, the structural configurations documented in the Valsorda valley constrain the orientation of the principal stresses into sub-horizontal  $\sigma_1$  (NNW-SSE) and  $\sigma_2$ , and sub-vertical  $\sigma_3$ .<sup>290</sup>

The movement of these faults and associated tectonic stylolites correspond to NNW-SSE compression. On the other side, at the flat-topped Latemar, on the sub-horizontal (pavement) outcrops, fractures also form conjugate patterns, and in most cases, exhibiting dextral and sinistral displacements (Figs. 4 and 5). These fractures documented from the field and drone images highlight the NNW-SSE, NE-SW and ENE-WSW orientations as the dominant fracture set and record an approximately NE-SW compression (Figs. 4 and 5b).

The measured bedding-perpendicular sub-vertical stylolites show the primary orientation of WNW-ESE and strike perpendicular to the bisection planes of the observed conjugate systems<sup>300</sup>

(Figs. 3 and 4c). These bedding-perpendicular stylolites are treated as compressional stress-field indicators, striking perpendicular to the maximum stress.

*Figure 3. Sub-vertical outcrops exposed at the base of the Latemar. a, b4 and b5) show reverse conjugate fault with low-angle SSE - and ENE dipping fractures and a horizontal intersection. In addition, (c2) show reactivated stylolites, which acted as a fluid-flow conduit. The stereoplots show bedding-perpendicular stylolites and σ1 striking approximately WSW-ENE and NW-SE to NNW-SSE direction, respectively.*

*Lastly, Overall, three deformation phases with sub-horizontal σ1 striking in different orientations were observed and documented in this field area. The reverse fault and strike-slip stress regimes with sub-horizontal σ1 striking approximately NW-SE to NNW-SSE and NE-SW, respectively (Figs. 3 through 5), and a later compressive deformation stage with a stress regime showing N-S sub-horizontal σ1. A later compressive deformation stage with a stress regime showing N-S sub-horizontal σ1 was also observed.* These stress regimes and orientations correlate well to the far-field sub-horizontal σ1 observed for the Latemar mountains during the Middle to Late Triassic and Neogene times, representing the subsidence – and Alpine deformation stages, respectively (Boro et al., 2013, 2014; Hardebol et al., 2015; Igbokwe et al., 2022). The orientations of these stress regimes, approximately NW-SE and N-S, were applied as the boundary loading directions defined in the modelling work presented here. The magnitudes of these stress regimes were assumed (i.e., idealistic; see section 3.4 for the rationale) to describe the impact of stress on the permeability of the carbonate rocks. Also, during the modelling, we assumed that fractures did not grow and no new fractures were formed, and all fractures, including stylolites, could be re-used as fluid-flow conduits.

*Although stylolite tends to hinder fluid flow (Boersma et al., 2019), observations in figures 3c and 4 and 5 show reactivated stylolites, which acted as a fluid-flow conduit. We refer the reader to Igbokwe et al. (2022), another paper as part of this project, where fractures and the stylolites were analyzed for their geometry, kinematics, and topology to delineate crosscutting relationships and the accompanying stress directions at the Latemar Carbonate Platform (N. Italy). they can enhance fluid movement.*



Figure 4. Sub-horizontal (pavement) outcrop exposed at the flat-topped Latemar. a and b) show high-resolution 2D outcrop orthorectified photograph and digitized (interpreted) fracture map showing reactivated styloites, which acted as a fluid-flow conduit. C) The stereoplots show  $\sigma_1$  striking approximately NW-SE to NNW-SSE direction.


### 3.1.1. Outcrop acquisition and digitisation

Fracture network patterns were acquired from the outcropping carbonate rocks, focussing on the sub-horizontal pavements of the flat-topped Latemar interior (Figs. 4 and 5a). The acquired images were processed using Agisoft Metashape® and converted into georeferenced digital 355 outcrop models using photogrammetry.

The observed fractures were interpreted and digitised using ArcGIS 10.5™ software, where fractures were traced and digitised with a polyline interpretation tool. Structural data such as length, orientation and spacing were computed for each polyline. The high quality of the drone images allows interpretation of thousands of potential fractures. Over 2,000 fractures were 360 documented, albeit with limited sampling and truncation artefacts.

Within the 2-D digitised fracture network of the sub-horizontal outcrop, five representative fracture network areas, referred to as fracture sub-structures (FSS), were selected and used to model stress heterogeneity and permeability. The dimensions of the individual FSS are approximately 2 x 2 m. For each FSS, the fracture length distributions and frequencies were 365 fetched from the digitised fractures and plotted in a histogram chart. The FSS areas are located in the same structural domain but display fracture networks with different spatial distributions.

Further, fracture networks were commonly sub-vertical, splaying, curving, intersecting, and tipping adjacent to other fractures.

-Locally, the fractures are orthogonal and/or in a conjugate pattern with a small conjugate angle.

~~Slight modifications and/or extrapolations of the fracture's original pattern were implemented to maintain the fracture topological connectivity.~~

### 3.3.3.2. Deformation of the fractured rock

In the following, the equilibrium equation for modelling the deformation of the rock matrix and fractures is presented as follows:

$$\nabla \sigma + \mathbf{b} = 0 \quad (1)$$

where  $\sigma$  is the stress tensor and  $\mathbf{b}$  is the body force vector.

In the 2D computational domain, the rock matrix is discretised with unstructured triangular elements. Using the linear elastic model, the deformation of the rock matrix is calculated. To capture the variation of the separation and slip of the fracture, a zero-thickness 380 interface element is introduced to represent the displacement gap on the fracture (Fig. 6a).

Taking into account the closure and frictional behaviour of fractures, the contact condition is imposed on the fracture surfaces by satisfying the standard Karush-Kuhn-Tucker condition in the normal direction (Oliver et al., 2008) as follows:

$$t_N^c = \begin{cases} 0 & \llbracket u_n \rrbracket \geq 0 \\ K_n \llbracket u_n \rrbracket & \llbracket u_n \rrbracket 

**Figure 5. Original outcrop model with interpreted fractures, from where the five FSS were extracted.**  
a) High-resolution 2D outcrop orthorectified photograph. b) Digitized sub-horizontal (pavement) in the Latmar with the position of the five outcrop windows, including the stereoplot of more than 2000 fracture orientations. c) The five fracture models. d) Length distribution of the five fracture models. e) Fracture geometry, including density and spacing values of the five outcrop models.

**3.4.3.3. Effective permeability of the fractured rock**

The effective permeability  $k_{eff}$  for a fractured rock was estimated by conducting a single-phase flow simulation in two perpendicular directions, using a computational homogenisation (Leonhart et al., 2017) for an unperturbed and a perturbed fracture state in uncoupled hydromechanics processes. The permeability tensors of the deformed carbonates were obtained 410 as

$$k_{eff} = \frac{\mu L}{m(\bar{p}^1 - \bar{p}^2)} \sum_{i=1}^m \mathbf{q}^i \cdot \mathbf{e} \quad (4)$$

$\bar{p}^1$  and  $\bar{p}^2$  are fluid pressure ( $\bar{p}^1 \neq \bar{p}^2$ ).  $\mathbf{q}$  is the fluid flow rate through the fractured rock.  $m$  is the number of Gauss points.  $L$  is the length of the domain of the numerical model. The dynamic viscosity  $\mu = 1 \times 10^{-9}$  MPa.s. When the pressures  $\bar{p}^1$  and  $\bar{p}^2$  are applied at the left and right boundaries and zero fluid flow conditions ( $\bar{q} = 0$ ) are applied at the top and bottom boundaries, the  $k_{eff}$  in the x-direction (Fig. 6b) is expressed as  $k_{xx}$  and the  $k_{eff}$  in the y-direction as  $k_{xy}$ . In contrast, when  $\bar{p}^1$  and  $\bar{p}^2$  are applied at the top and bottom boundaries and zero fluid flow conditions ( $\bar{q} = 0$ ) are applied at the left and right boundaries,  $k_{eff}$  in the y-direction is considered to be  $k_{yy}$ .

The fluid flow is assumed to be in steady-state condition for calculating the effective permeability. The flow in fractures and the rock matrix are both considered. To simulate and model the fluid flow, the unified pipe-interface element model proposed by Yan et al. (2021, 2022) and Darcy's law were employed (Fig. 7a), respectively. For a steady saturated incompressible fluid flow, the mass conservation equation in each node is expressed as (Ren et al. 2017):

$$\sum_{n_i=1}^{N_i} K_{n_i} \Delta p_{n_i} = q_{s_i} \quad (5)$$

Here, subscript  $N_i$  denotes the number of pipes connecting to the node.

$q_{s_i}$  is the source term of the node  $i$ , and  $K$  is the conductance coefficient of the pipe. The detailed derivation of the conductance coefficient for fracture pipe and rock matrix pipe can be found in Yan et al. (2020).

For the fracture with an aperture  $\llbracket u_n \rrbracket$ , the conductance coefficient  $K_{fp(i,j)}$  could be calculated as:

$$K_{fp(i,j)}^{2D} = \frac{\llbracket u_n \rrbracket^3}{12\mu l_{ij}} \quad (6)$$

where  $l_{ij}$  is the length of the fracture segments.

As shown in figure 7b, each line between two nodes represents a pipe approximating the fluid flow through the rock matrix. The circumcenter of the triangle is used to derive the parameters of pipes forming a triangular mesh. The line linking the circumcenter and the mid-point of each edge is perpendicular to the edge, and these perpendicular bisectors divide the triangle into three parts. The matrix pipe  $ij$  represents the fluid transfer between domain ***o-c-i-a*** and domain ***o-a-j-b*** through their common face ***oa***, and its flow rate is denoted as  $q_{oa}$ . By making the flow rate  $q_{ij}$  in pipe  $mp(i,j)$  equivalent to the flow rate in the matrix  $q_{oa}$ , the conductance coefficient  $K_{mp(i,j)}$  could be calculated as:

$$K_{mp(i,j)}^{2D} = \frac{l_{0a} k_m}{l_{ij} \mu} \quad (7)$$

Figure 6. (a) Representation of a fractured rock using triangle elements and embedded zero-thickness interface elements in the solid model. (b) Calculation model of the effective permeability

We have adopted the matrix permeability value of  $2 \times 10^{-15} \text{ m}^2$  in Whitaker et al. (2014)

drawn from the forward-coupled modelling of the sedimentary and diagenetic evolution in the
 Latemar carbonate buildup. We acknowledge that in some places, due to diverse (diagenetic) alterations (dissolution/ precipitation), the matrix permeability of different lithologies carbonate rocks can be very high, reaching up to 5 D, as noted in the simulated permeability values in some carbonate lithologies in Latemar (Whitaker et al., 2014). Also, previous studies in Latemar recorded matrix permeability in the Latemar margin and interior in the order of 1-30 mD for low-permeable end-member and up to 300 mD for high-permeable member (Whitaker et al., 2014; Hardebol et al., 2015). But, because carbonate rocks, in general, have very low matrix permeability with average values ranging from 1 to  $4 \times 10^{-15} \text{ m}^2$  (Hardebol et al., 2015), we have constrained our values to 2 mD, which is only indicative approximation.

Meaning that changing the matrix permeability value will impact the final result. However, such a sensitivity analysis is beyond the scope of this study.

Figure 7. (a) Scheme of matrix pipes for fluid flow; (b) Pipe equivalence of rock matrix

### 3.5.3.4. Boundary conditions

The boundary conditions are defined by two stages of far-field sub-horizontal  $\sigma_1$  (stress) loading scenarios (Fig. 8), assumed to have acted on the studied carbonates forming the 500 Latemar buildup during the subsidence-related and Alpine deformation stages. These tectonic episodes are constrained to the NW-SE and N-S shortening (compressive) directions, which is parallel to the current-day stress orientation that has been estimated from seismic and active faults data in Italy and the central Mediterranean (Montone et al., 2004; Pierdominic & Heidbach, 2012; Heidbach et al., 2018). The applied differential stresses are set at 50 MPa ( $S_H = 50$  MPa,  $S_h = 0$  MPa; with an assumed maximum magnitude) of 50 and 160 MPa for the 505 subsidence-related deformation stage in the NW-SE direction, and 160 MPa ( $S_H = 160$  MPa,  $S_h = 0$  MPa; maximum magnitude) for the Alpine deformation stage in the N-S directions, respectively. These values are close in range with the magnitude values for modelling

compressive settings in carbonate platforms within similar geodynamics settings (e.g., Boresma et al. 2019). Further, we implement Poisson's ratio ( $\nu$ ) of 0.30 and Young's modulus ( $E$ ) of 25 GPa (see Table 1 for detailed parameters), and for carbonate rocks, these values are achievable under subsurface conditions (Goodman, 1989; Bertotti et al., 2017). It should be noted that the influence of the overburden stresses was not taken into account, as the numerical modelling is in the 2-D. Thus, limiting the ability to consider the impact of lateral expansion on the fracture domain. In addition, the applied current-day stress orientation could be different to the paleostress field during the formation of most parts of the Latemar platform. Therefore, we acknowledge that changing the applied stress directions and magnitudes will alter the modelled apertures and the final calculation of the effective permeability. Nevertheless, such a sensitivity analysis is outside the purview of this study.

KRATOS, an open-source multi-physic Finite Element software (Dadvand et al., 2013), was used to conduct the 2-D linear-elastic mechanical modelling by applying loads and/or displacement boundary conditions to the models (Fig. 2b; Phase 2.2). Due to the highly nonlinear behaviour of the fracture model, a linear elastic 2D plain strain constitutive model was used for the rock matrix to keep the simplicity in the simulation. The constitutive parameter values are Youngs modulus 25 GPa and Poisson ration 0.30 (see Table 1 for detailed parameters). A 2-D plane strain linear elastic isotropic matrix was assumed and used to treat all modelled fracture planes equally, especially on the  $z$ -axis.

The stress distributions were analysed using triangular and interface elements corresponding to the rock matrix and fractures. The effects of the far-field stresses on the overall model and the relative displacement of the deformed and undeformed element geometries were simulated. Given that all the representative FSS are sub-horizontal, the origin of the  $x$  and  $y$ -axis is placed at the plane's centre. In other words, the centre of the FSS is to be fixed in both

the x- and the y-axis so that translation and rotation are prevented. The loads are applied on the outer boundaries of FSS, as shown in figures 2 and 8.

**Figure 8. Quasi-static loading scenarios of the two tectonic stresses.** a) The first stage loading reflects the subsidence deformation stress from the NW-SE direction and has a maximum load of 50 MPa. b) The second stage loading, representing the Alpine deformation stress from the N-S direction, is applied (added), while the loading of 50 MPa is maintained until a maximum of 160 MPa is reached. c) The plot of loads increased in discrete *Pseudo* Time Steps (PTS). The first stage loading of 50 MPa corresponds to *line a-b-c* (blue curve), whereas the second stage loading of 160 MPa corresponds to *line d-e-f* (orange curve). The grey bar indicates that some stresses during the first loading were still active and became more significant during the second loading scenario.

The loading of the first tectonic episode, subsidence deformation, represented as first-stage loading, was applied from the NW-SE direction. This loading was increased in load steps, representing Pseudo Time Steps (PST), analogues to quasi-static loading, until a maximum magnitude of 50 MPa was reached at PTS1 (Fig. 8a, c). After that, the loading of the second tectonic stage, Alpine compression, known as second stage loading, was added (or applied) 550 from the N-S direction, while the compressive loading corresponding to the first stage was active and/or maintained at 50 MPa. Analogous to stage 1 loading, the load values were increased in loading steps until a maximum value of 160 MPa in PTS 2 was attained (Figs. 2b, 8; Phase 2.2.1). Lines *a-b-c* (blue curve) and *d-e-f* (orange curve) depict these loading stages, and the grey bar in the graph of the Load (MPa) in pseudo time steps (Fig. 8c) represents the 555 transition zone between the two loading scenarios. These applied boundary loadings are analogous to in-situ stresses, ~~which are feasible magnitudes for modelling compressive settings (Heidbach et al., 2018)~~.

The essential model parameters are listed in Table 1. The modelled local state stresses and slips were used to calculate the local stress-induced aperture and their distributions, even as individual fractures open differently.

#### 4. Numerical results

Figure 2 depicts the workflow used to model the aperture and permeability distributions through an outcropping network of fractures. Extracting five selected outcrops (e.g., FSS1 – FSS5) from the fractured pavement, outcrop models developed demonstrate in more detail the impact of the mechanical deformation (loading), change in stress regimes and fracture geometry on modelled aperture and effective permeability. In addition, the flow anisotropy function is evaluated in response to the loading scenarios and orientation.

Figure 5b shows embedded five representative FSSs, each containing sufficient fracture heterogeneities to provide a representative value for properties such as fracture densities, 570 spacing, connectivity, etc. Geometrical analysis of the outcrops indicates a varied average fracture length and density among the FSS. For example, the average length of the selected FSS ranged from 0.5 to 3.5 m, whereas fracture density (P20) was comparable in FSS1, FSS2, and FSS3 ( $10.5 \text{ m}^{-2}$ ,  $21 \text{ m}^{-2}$ ,  $13.3 \text{ m}^{-2}$ ), but dissimilar in FSS4 and FSS5 ( $62.3 \text{ m}^{-2}$  and  $42.5 \text{ m}^{-2}$ ; Fig. 5d, e). The average fracture spacing (in each FSS) for each fracture orientation ranges 575 between 0.7, 0.72 and 0.4 m for NNW-SSE, NE-SW, and ENE-WSW, respectively (Fig. 5e). The dihedral angles between the different fracture orientations measure between 18 and 60°.

#### 4.1. Stress orientation effects and aperture distributions

The local distributions of stresses and the computed aperture distributions from the local state of stress are presented. The effects of changing stress regimes, from NW-SE and N-S directed 580 stress cases, for each FSS domain are shown in figure 9, demonstrating the distributions of Von-Mises stress in all the FSS due to applied loading conditions. At the onset of the subsidence stage, a uniform distribution of the local stress state at PTS 0.01 is observed. This uniform distribution of local stress increases in discrete steps gradually until PTS 1 is reached. At PTS 2 (Alpine deformation stage), the increasing loading in the N-S direction shows the 585 disperse of the uniform stress distribution with noticeable fluctuations in magnitudes, pointing

to the varied change in the local state of stress caused by the heterogeneity of fractures (Fig. 9).

**Figure 9. Von-Mises stress distribution (Pa) in the fractured rock domain (FSS 1 through FSS 5) under different loading conditions and orientations at Pseudo Time Steps (0.01, 1.0, and 2.0). Note that the deformation scale is x10. The stress colour spectra are the same for all figures.**

Figure 10 depicts the displacement magnitude distribution in each FSS when the loading conditions are at PTS 2. The observations in figures 9 and 10 show how changes in the magnitude of sub-horizontal far-field stresses cause varied deformation and/or influence the individual fractured rock domain, the FSS. These changes, particularly in figure 10, are, 595 however, quantified for each FSS, ranging from 0 - 0.012 m (FSS1), 0 - 0.079 m (FSS2), 0 - 0.080 m (FSS3), 0 - 0.034 m (FSS4), and 0 - 0.015 m (FSS5). It is observed that the total displacement magnitude in the model increases with increased fracture density and connectivity. For example, FSS2 and FSS3 show a slight increase in the displacement magnitude due to increased fracture density and intersection (Figs. 5e and 10).

**Figure 10. Distribution of actual displacement magnitude under the two tectonic stress conditions for FSS1 through FSS5, when the Time Step is 2. Units are in meters, and deformations are slightly magnified. Notice the varied displacements across the FSSs**

Depending on the magnitude and orientation of the loading scenarios, fractures open, close and/or shear. The fracture openings (apertures) are constrained in varied sizes, varying between 605 2 and 60 mm, and their overall distributions against the number of fractures are shown in figure 11a. For each FSS, the numbers of fractures are plotted against their aperture values, and these plots show distinct heterogeneous apertures of variable sizes within each FSS (Fig. 11a). Quantifying the changes in aperture as a function of increasing stress and changing stress orientations for each FSS, figure 11b shows the general trend of an average aperture, which is 610 changing with the changing loading condition, inside each FSS. For instance, in each FSS, a rapid increase in aperture value was observed as the first loading in the NW-SE direction increased, and this value peaked at PTS 1 (at 50 MPa). The trend of aperture values changes at the second loading stage in the N-S direction. For example, in this case, FSS 1, 4 and 5, the aperture values initially decrease rapidly until PTS 1.3 is reached. After that, aperture trends 615 remained constant for FSS 4 and 5, whereas FSS 1, 2, and 3 aperture values gradually increased until reaching their peak values at PTS 2 (160 MPa; Fig. 11b).

|                                      |        | Average data | Data range  |
|--------------------------------------|--------|--------------|-------------|
| Surface Tension ( $\text{Jm}^{-2}$ ) | $Y$    | 0.27         | 0.27        |
| Young's Modulus (GPa)                | $E$    | 25           | 15 – 45     |
| Poisson's ratio                      | $\nu$  | 0.3          | 0.25 – 0.3  |
| Rock density ( $\text{Kgm}^{-3}$ )   | $\rho$ | 2200         | 2000 – 2700 |
| Tectonic Stress (MPa)                |        | 10           | 0 – 35      |
| Frictional co-efficient              |        | 0            | 0           |

**Table 1.** Essential model parameters applied for stress calculations. Adapted from Bertotti et al. (2017)

#### 4.2. Effective permeability as a function of loading

The effective permeability in each FSS is presented as a function of the NW-SE and N-S loading scenarios. Fracture apertures mechanically generated were applied to the finite element models, and by running the single-phase flow simulations, the effective permeabilities for each loading condition (stress orientation) were obtained. For a  $2 \times 10^{-15} \text{ m}^2$  matrix permeability, the flow is calculated based on the fracture densities and orientations in all the FSS. The flow paths 625 are linked to the areas where there is a high number of fractures. Figure 12 shows a nearly homogenous long-term steady-state fluid pressure distribution and gradients over the fractured rock domain (FSS) at the quasi-static loading at PTS 2 in the x- and y-directions. This result

points to the steady-state condition of the pressure field, serving as the base from which the effective permeabilities of all the selected FSS were computed.

In addition, using the parameters in Table 1 and considering the initial fracture aperture to be zero, figure 13 shows the computed effective permeabilities in both x and y directions (the red curve =  $k_{xx}$  and the black curve =  $k_{yy}$ ). Also, the components of the permeability tensor for each loading condition are shown in Table 2.

| Time Steps     | Effective Permeability ( $K_{\text{eff}}$ ) |             |                 |
|----------------|---------------------------------------------|-------------|-----------------|
|                | $K \times 10^{-15} (\text{m}^2)$            |             |                 |
|                | Matrix permeability = 2.00                  |             |                 |
|                | $K_{xx}$                                    | $K_{yy}$    | $K_{xx}/K_{yy}$ |
| <b>FSS 1</b>   | 1                                           | 2.14        | 2.20            |
| <b>FSS 2</b>   | 1                                           | 2.30        | 2.18            |
| <b>FSS 3</b>   | 1                                           | 3.10        | 2.60            |
| <b>FSS 4</b>   | 1                                           | 3.40        | 3.70            |
| <b>FSS 5</b>   | 1                                           | 2.80        | 2.54            |
| <b>Average</b> | <b>1</b>                                    | <b>2.75</b> | <b>2.64</b>     |
| <b>FSS 1</b>   | 2                                           | 2.28        | 2.70            |
| <b>FSS 2</b>   | 2                                           | 2.42        | 2.56            |
| <b>FSS 3</b>   | 2                                           | 3.28        | 3.90            |
| <b>FSS 4</b>   | 2                                           | 3.50        | 4.20            |
| <b>FSS 5</b>   | 2                                           | 2.36        | 2.84            |
| <b>Average</b> | <b>2</b>                                    | <b>2.77</b> | <b>3.24</b>     |
|                |                                             |             | <b>0.85</b>     |

645 *Table 2. Effective permeabilities, including permeability Tensor obtained at different loading scenarios*

The results from the first loading stage (PTS 1, representing the subsidence deformation stage from the NW-SE direction) show a gradual increase in permeability beyond the matrix 650 permeability within each FSS (Fig. 13). The increased permeability values correspond to increased aperture values in figure 11b, pointing to the initial opening of most fractures parallel to the NW-SE direction. On the other side, for the second loading stage (PTS 2), for which the loading is applied from the N-S while the first loading stage is active, the permeability values are seen to increase in varying degrees. While the close fractures have reduced aperture and 655 permeability values, opened fractures exhibit increased permeability values (Figs. 11b and 13). Further, in both loading scenarios, a general observation shows that the permeability values are variable in each FSS, pointing to the random distribution of fractures, varied densities (Fig. 5) and varied aperture values (Fig. 11a). For instance, in loading stage 1 from the NW-SE orientation, the results in the permeability plots show both the vertical and horizontal 660 permeabilities increase gradually in FSS 1 through FSS 5 (Fig. 13).

Figure 11. a) Distribution of fracture aperture under the two horizontal stress orientations. b) The relationship between aperture and mechanical loading conditions. Note that the Time is analogous to quasi-static loading conditions.

Although FSS 1, FSS 2 and FSS 5 have their permeability values (in both x- and y-direction) close to the matrix permeability, FSS 3 and FSS 4 only show permeability values increased by 55 and 70 % in comparison to the matrix permeability (Table 2). Nevertheless, with the second loading scenario from the N-S orientation, the  $k_{xx}$  and  $k_{yy}$  maintained a

relatively steady permeability value with a slight increase in FSS 3 through FSS 5. This is

695 unlike FSS 1 and FSS 2, which recorded a sharp increase (a jump) in both  $k_{xx}$  and  $k_{yy}$ .

**Figure 12.** Contours of the fluid pressure gradient (Pa), obtained for a fluid pore pressure  $p$  at Time Step 2 (see Fig. 9 for the loading conditions), in the x- and y- directions. The matrix permeability is given as  $2 \times 10^{-15} \text{ m}^2$

Quantitatively, the average permeability for all the FSS is given as  $2.75 \times 10^{-15} \text{ m}^2$  and

$2.64 \times 10^{-15} \text{ m}^2$  in the x- and y-direction for PTS 1, and  $2.77 \times 10^{-15} \text{ m}^2$  and  $3.24 \times 10^{-15} \text{ m}^2$  in the x- and y-direction for PTS 2, respectively. When compared to the matrix permeability, these average permeability values,  $k_{xx}$  and  $k_{yy}$ , increased by 37.5 and 32% for PTS 1 (at the maximum magnitude of 50 MPa in the subsidence stage) and 38.5 and 62% for PTS 2 (at the maximum magnitude of 160 MPa during the Alpine deformation stage).

**5. Interpretation and Discussion**

The results of the structural analysis and geomechanical (numerical) simulations show how stress regime change can impact the permeability of a naturally fractured carbonate rock. It also highlights the effect of changing stress magnitude and orientation on apertures at the fracture network scale quantified in the two-dimensional network. In addition, this process can 725 be the first step toward using fractures (and faults) as a flow medium. This is especially true

for the upscaling and larger-scale numerical simulations important for fluid flow in geothermal and hydrocarbon systems.

**Figure 13.** The Effective permeability values are plotted against the loading conditions for FSS1 through FSS5. The FSS 1 through 3 show significant “jump in permeability values” at Pseudo Time Step 1 when the second quasi-static loading commences.

### 5.1. Subsidence- and Alpine deformation: establishing a realistic loading condition for the Latemar buildup

Deformation in the carbonates of the Latemar buildup caused the development of different sets of fracture networks, including veins. As documented, these fracture networks formed during 735 syn-sedimentary extension tectonics in the Middle Triassic, characterised by wholesale

subsidence deformation and faulting tectonics (Boro et al., 2013; Preto et al., 2011). Other studies (e.g., Goldhammer and Harris, 1989; Emmerich et al., 2005) suggest that the subsidence deformation in Latemar may have continued until the Late Triassic or Early Jurassic times. Subsequently, the Latemar buildup was affected by the Alpine deformation between the Late 740 Paleogene and Early Neogene times. These tectonic episodes significantly impacted the hydraulic properties of fracture networks at the Latemar.

Given the complexities of the modelling work presented here, the studied fracture networks were assumed to have formed earlier, prior to the major tectonic episodes, as documented in Preto et al. (2011). Although both tectonic events may have developed new fractures, the 745 modelled tectonic stresses in this study did not allow for further development of fractures but for fractures to open, shear and/or close, consequent on the magnitude and orientation of the stresses.

Our structural analysis results (Figs. 3 and 4) reveal the orientation of the principal stress fields affecting the fractures at the subsidence and extension tectonic stage, which is roughly 750 NW-SE to NNW-SSE trending sub-horizontal  $\sigma_1$  and a sub-vertical  $\sigma_3$ . This compression is believed to have occurred relatively shallow or at an intermediate burial depth. Indirect evidence for this notion comes from the understanding that, in the presence of sub-horizontal tectonic stress, a sub-vertical position of  $\sigma_3$  is compatible with low-angle reverse faults (or structures), forming at shallow to intermediate burial depth (e.g., Fig. 3; Bisdom et al., 2016c; 755 Bertotti et al., 2020). This means that the overburden stress is the resultant far-field stress at the subsidence stage with an approximate NW-SE orientation and an assumed magnitude of 50 MPa.

On the other side, the main Alpine deformation in the Dolomite Mountain Belt (including the study area) involved maximum compressive stress ( $\sigma_1$ ) orientated approximately N-S with 760 a magnitude estimated at 160 MPa (Peacock, 2009; Abbà et al., 2018). Because the observed correlation between the fractures (and their networks) and stress fields is largely clear, the magnitude of the two sub-horizontal  $\sigma_1$  served as the input parameters for the numerical mechanical modelling phases, i.e., the mechanical loads used during the mechanical FE model setup. These mechanical loads captured the realistic compressive boundary condition, 765 reflecting the tectonic episodes of the study area, unlike previous studies (e.g., Zhang and Sanderson, 1996) and are essential for the realistic computation of effective permeabilities, even at the subsurface.

Figure 8c reveals that the deformation caused by the first stress field (NW-SE shortening direction at subsidence stage) was still active, as the second stress field (N-S sub-horizontal

$\sigma_1$ ; Alpine deformation) was added. The gap (the grey bar Fig. 8c) between the two stresses in our model signifies that some stresses in the horizontal dimension due to subsidence deformation are present (active) at the Alpine deformation stage when they increase and become more extensive in the N-S direction. This is in contrast with previous studies, for example, Stephansson et al. (1991) and Yale (2003), where modelling results (or model setup) 775 utilise a homogenous stress regime that does not account for all other stress impacts around the fracture network. Each effect remains independent when all the stress regime changes are considered (Agheshlui et al., 2018; Bisdom et al., 2017). That is, the impact of the first or second stress field, as the case may be, is not kept active when the third or fourth stress field is implemented. Unique to our study, the orientation and magnitude of all the stress regimes, 780 reflecting the major tectonic event over a geological period, are kept active and accounted for during the geomechanical modelling. Therefore, this makes the impact of stress irreversible in a given rock domain, and the consequence of the absence of these stresses can introduce considerable error when calculating the effective permeability, especially at the subsurface reservoir scale. This argument follows the proponent of representing the realistic tectonic 785 events affecting fracture networks on the geomechanical models. Thus, proposing that modelling the impact of different tectonic episodes (in our case, the subsidence- and Alpine deformation) on a given rock domain (FSS, in the Latemar buildup) over a geologic time (Triassic to Neogene) will significantly reduce uncertainties in computing the apertures and permeabilities, even at the reservoir scale.

Based on this, the loading conditions and the calculated permeabilities presented in figures 8 and 13 may reflect close occurrences of what is obtainable at the subsurface of a natural carbonate reservoir.

## 5.2. The link between heterogeneous aperture, fracture geometry and the impact of stress and effective permeability

As a rule, the geomechanical models and flow simulations in fractured rocks have always depended on (i) the subsurface datasets, which are typically expensive, albeit with uncertainty ranges (Bourbiaux et al., 2002; 2005), and (ii) stochastic datasets (Khodaei et al., 2021; Timothy & Meschke, 2016), which are not realistic when considering the behaviour of a fracture network in a natural reservoir. In contrast to these studies, the presented investigation 800 solely uses the outcropping network geometry as input for geomechanical and flow models, not considering the outcropping apertures. Instead, the geomechanical aperture sizes were

modelled in a representative outcrop fracture network using computational homogenisation (FE mechanical modelling) for unperturbed and perturbed fracture states. The natural fracture system is perturbed by applying mechanical load, analogous to in-situ stress.

The combination of outcrop fracture geometries, mechanical loading and aperture distribution results in models that are more representative of fractured reservoir permeability compared to analogue studies that use apertures of exhumed barren fractures or assume a constant aperture for the whole fracture network (Bisdom et al., 2017; Makedonska et al., 2016). Like most fractured reservoir models, the aperture is assumed constant per fracture or 810 even per fracture set because generating a reservoir scale 3-D fracture model with mechanically controlled heterogeneous aperture distributions can be complex (Geiger and Matthäi, 2014). However, studies by Jonoud and Jackson (2008) and Cottreau et al. (2010) have given an upscaling alternative through arithmetic or harmonic averaging of the explicit fracture permeability model calculated per fracture node. That is, the mechanically controlled 815 heterogeneous aperture, like the mechanically generated aperture in figure 11, can easily be upscaled to serve a more representative reservoir scale fracture model.

Further, fracture networks, including intensely fractured zones, are believed to significantly influence the effective permeability and fluid flow patterns in a naturally fractured reservoir, particularly in tight carbonate reservoirs. These network areas show high porosities and 820 permeabilities relative to the surrounding host rock (Bruhn et al., 2017; Matthäi, 2003). It is a widely held view that the extent to which the fracture networks impact flow lies in the fracture's structural arrangement and geometry, such as fracture orientations, spacing, and length (Bisdom et al., 2016a, c; Hardebol et al., 2015; Olson, 2003; Scholz, 2010). However, these 825 views are limited to models that quantify aperture based on fracture length and spacing relations (Olson, 2003; Scholz, 2010). The results from our models (Figs. 11 and 13) indicate that aperture and effective permeability are not easily related to fracture geometrical parameters such as length or spacing. This is especially true because the linear functions of fracture lengths and spacing have little or no effect on the mechanically generated aperture distribution (Bisdom et al., 2016c), controlled by fracture orientation and the impact of stress.

Given these results, we suggest that the distribution of aperture and permeability (in the presented models) is influenced by the impact, magnitude and direction of the stresses, fracture orientation and shear displacement (Figs. 11 and 13). For the stress-induced aperture, the mechanical load opens individual fractures orthogonal to the direction of loading at different rates and simultaneously closes the fractures parallel to the loading direction (Heffer & 835 Koutsabeloulis, 1995). Therefore, in a favourable orientation, that is, the direction with

significant stress components (of mechanical loading), which in our case is NW-SE for the subsidence deformation, the results show a considerable increase in permeability as the load increases (Fig. 11) before reaching a peak at PTS 1. The increased permeability – both in the x- and y-direction – exceeds the matrix permeability within all the FSS by 37.5 and 32%, 840 indicating a structural change. On the other side, permeability continues in an upward trajectory with varying degrees of increased values (Fig. 13) in the direction of more significant stress components (N-S) at the Alpine deformation stage for all the FSS until a maximum loading value is reached. In this case, the average increased permeability is beyond the matrix permeability by up to 62%, in some of the FSS, especially in the y-direction. These suggest a 845 substantial structural change of the fractures within the fracture network, further indicated by the varied aperture distribution.

The evolution of permeability values in x- and y-directions within the FSS indicates that the values of the permeability distribution are variable. These variabilities are linked to the varied fracture densities and aperture values (Fig. 5). In fractured reservoirs, apertures of 850 natural fractures are highly heterogeneous and can contribute to the induced anisotropy in the permeability (Makedonska et al., 2016). In our case, for instance, FSS 1, FSS 3, and FSS 5 significantly differ in the evolution of the permeability values in the x- and y-directions and distinctly show induced anisotropy in the permeability within their rock domains (Fig. 13). This is demonstrated by the differences in the permeability values in x- and y-directions. The 855 higher the difference in the permeability values of the x-and y-direction, the more significant the induced anisotropy. These differences are linked to the fracture connectivity and/or distributions (Fig. 5c, d) within each FSS.

Conversely, FSS 2 and FSS 4 show isotropy in permeability, as the differences in the permeability values of x- and y- directions are relatively small compared to those in FSS 1, 860 FSS 3, and FSS 5. These behaviours can be explained as follows: initially, during the subsidence stage, when the far-field stress (or compressive loading) is in the NW-SE direction, the fractures parallel to this direction open while the fractures orthogonal to the NW-SE direction close. The opening of the NW-SE-directed parallel fractures increases the overall permeability in all FSS (Fig. 13). However, at PTS 1, the second set of loading scenarios is 865 gradually applied from the N-S direction at the onset of Alpine deformation. While the loading from the first stage is kept constant, the overall permeability at this stage continues with varying degrees of increase until PTS 2. This implies that the new loading opens the fractures parallel to the N-S direction and closes the fracture orientation in the E-W direction (Fig. 13). Hence, the second loading stage, in principle, opens the fractures that were previously closed by the

870 first loading stage. This means that many fractures opened, resulting in larger permeability values. There is also a documented sharp jump (kink style structure) in permeability values plotted for FSS 1, FSS 2 and FSS 3 (Fig. 13) at the beginning of the second loading stage. This sharp increase in the permeabilities may imply that fractures oriented slightly in the NE-SW and/or NW-SE directions (Fig. 5c) may have been opened by the N-S loading condition.

In addition, fracture density and/or the number of fractures in each rock domain, FSS, may have played a considerable role in the sharp spike in permeability values. For example, in FSS 1 and FSS 2, where the quick jump in permeability values is predominant, fracture density is relatively lower than in FSS 4 and FSS 5, with high fracture density and no visible sharp increase. These explain the jump and the increase in permeability during the second loading 880 stage. Recent studies have linked the fracture density, the number of fractures and the stress orientation with an overall increase in the permeability values (e.g., Bisdom et al., 2016b, c; Furtado et al., 2022). Although some fractures may have been sheared and eventually opened during loading, we believe that the shearing of these fractures has minimal or no impact on the overall permeability of the rock domains.

One interesting observation in this research investigation is that the two far-field stresses acting within a rock domain simultaneously with different magnitude and orientation (for example, at PTS 2, in our case) anisotropically stressed the fracture network, which does not necessarily cancel out each other in the mechanical FE modelling phases. However, these stresses create new effects on the overall aperture and permeability distributions linked to 890 fracture geometry, connectivity and density. In our case, there is a general increase in the overall aperture and permeability values, respectively (Figs. 11b and 13), contingent on subsurface energy storage in carbonates.

### 5.3. Implications

All modelled features are assumed to be reactivated fractures that control the fluid flow in the presented numerical workflow. Because of the effects of weathering and exhumation, the distinction between open fractures, veins, stylolites and shear fractures was not made. Although this assumption is widely used in reservoir modelling, complex features such as sub-vertical (tectonic) stylolites or partly cemented fractures can inhibit or enhance fluid flow (Bruna et al., 2019). Cemented fractures strongly constrain fluid flow, as cementation reduces the aperture of a fracture plane (Gale et al., 2010; Olson, 2007).

Although our study permits investigating the impact of stress regime change in detail, the results will broadly differ in complex and large-scale reservoirs since our model is orders of magnitude smaller than a subsurface reservoir. The studied fracture networks have relatively simple geometries on a small scale compared to what is usually observed in large-scale reservoirs. However, Matthai and Nick (2009) and Bisdom et al. (2016a) upscaled heterogeneous aperture distribution to a single averaged aperture, which provided the same permeability distribution. This means that averaging and upscaling the permeability of small-scale fracture models can accurately describe permeability when most fracture contributes to flow. Otherwise, the small-scale models may not be representative.

Another issue is ignoring the influence of the overburden stress on fracture aperture. Overburden stresses result in lateral expansion of rock layers, and in addition to the horizontal stresses, fracture apertures are strongly dependent on overburden stresses due to Poisson's effect (Agheshlui et al., 2018). Therefore, the numerical analysis in 2-D constrains the effects of lateral expansion of the fractured rock domain, which can displace the rock body and increase the fracture aperture. This means that the results presented in our model may only provide indicative approximations.

Further, better knowledge of stress conditions and elastic parameters aids in predicting realistic aperture descriptions for reservoir models. Studies have noted the impact of these parameters on modelled stress-induced aperture such that stress orientations and the magnitude of differential stress largely influenced the resulting aperture distribution (Agheshlui et al., 2018; Bisdom et al., 2016a). The presented model assumed the magnitudes of the two stress regimes following the calculated palaeostress in other field areas with the same geodynamic conditions. The values of these magnitudes are debatable and subjective. In addition, the arbitrary choice of the 2 x 2 m FSS may not guarantee that this is an actual representative sample for calculating permeability. Therefore, changing the magnitude of these stress values and increasing the FSS size can affect this study's overall aperture and permeability distributions. Still, the workability of the model remains stable and can compute any given matter.

Lastly, the off-diagonal components of the permeability tensor were not calculated, and the multi-field coupling methods, such as flipping the order of applied stresses, were not considered in this study. These, when considered, may affect the overall results and will be considered in detail in the forthcoming paper looking at the three-dimensional fracture network.

## 6. Conclusions

A detailed workflow is presented that applies a displacement-based linear elastic finite element method (FEM) to model the impact of stress regime change on the permeability of carbonate rock. Considering the tectonic episodes (Subsidence- and Alpine deformations) and outcropping fracture network at the Latemar, the stress-induced heterogeneous aperture distribution was generated in the selected Fracture Sub-structure (FSS). The impact of stress  
regime change on flow was quantified regarding varied aperture and effective permeability. Although the studied fracture networks have relatively simple geometries compared to what is ordinarily observed in the reservoir scale, this permits investigating the impact of individual stress magnitude and stress regime change on permeability.

First, we presented the structural analysis and interpretation of fracture network geometry in the study area, considering the two major tectonic episodes, the subsidence – and Alpine deformation. Their stresses were considered far-field stresses (compressive loading), which informed the boundary conditions (two loading stages) in the geomechanical FE modelling. The first stage of loading acts from the NW-SE direction, representing the subsidence deformation, and gradually reaches a maximum loading condition at 50 MPa in pseudo time  
steps (PTS 1), representing the load increment. In contrast, the second loading stage, the Alpine deformation, was superimposed on the first and gradually reached a maximum value of 160 MPa in PTS 2.

Second, it was shown that in the directions with less significant stress components (of mechanical loading), NW-SE, the overall permeability increased gradually exceeding the matrix permeability, both in the x- and y- direction, within all the FSS by 37.5 and 32%, respectively, as the load increased before reaching a peak at PTS 1. In addition, permeability increase continues with the upward trajectory in the direction with more significant stress components (N-S) for all the FSS as the load increases. The average increased permeability is beyond the matrix permeability by up to 62%, especially in the y-direction. Fracture density,  
the number of fractures and the stress orientation play a critical role in the overall increase in the permeability values.

Finally, we conclude that the two superimposed mechanical loadings, simultaneously acting within a rock domain, at a different magnitude and orientation anisotropically stressed the fracture network. These mechanical loadings do not necessarily cancel out each other in the mechanical FE modelling phases. However, these mechanical loads create new effects in the overall aperture and permeability distributions. The new effects include, but not limited to,

average increased permeability values beyond the matrix permeability by up to 62%, especially in the y-direction. With this increased average permeability values, fracture geometry, connectivity and density are implicated, and these can be complicated, depending on other

numerous factors, including temperature and fluid pressure, in large subsurface reservoirs.

However, these mechanical loads create new effects in the overall aperture and permeability distributions, which are linked to fracture geometry, connectivity and density and can be complicated, depending on numerous factors, including temperature and fluid pressure, in large subsurface reservoirs.

## 975 Author Contribution Statement

The first author named is the lead and corresponding author. All other authors are listed according to their contributions. We describe contributions to the paper using Contribution Roles Taxonomy (CRediT)<sup>1</sup> as follows:

**Onyedika Anthony Igbokwe:** Conceptualization, Methodology, Software, Formal analysis, Investigation, Data curation, Writing- Original draft preparation, Visualization, Project administration. **Jithender J. Timothy:** Data curation, Writing- Reviewing and Editing, Validation. **Ashwani Kumar:** Investigation, Software, Resources, Data curation. **Xiao Yan:** Software, Validation, Data curation, Formal analysis. **Mathias Mueller:** Resources, Data curation. **Alessandro Verdecchia:** Resources, Data curation. **Günther Meschke:** Supervision, 985 Project administration. **Adrian Immenhauser:** Supervision, Writing- Reviewing and Editing, Validation, Project administration.

## Competing interest

The contact author has declared that none of the authors has any competing interests.

## 990 Acknowledgements

The Petroleum Technology Development Fund (PTDF) Abuja, Nigeria, is thanked for sponsoring the doctoral fellowship of the first author (Grant number: OSS/PHD/IOA/843/16).

The authors thank G. Bertotti for discussing the tectonic significance of the fracture model, and N. Gründken and S. Schurr for their assistance during the fieldwork. Thanks to J. Dikachi-Igbokwe for her helpful comments, which helped improve this paper. Fieldwork was supported by grant DFG FOR 1644 to A. Immenhauser.

<sup>1</sup> Accessed from <https://www.elsevier.com/authors/policies-and-guidelines/credit-author-statement> (Sept. 20, 2023)

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
