# Peer review of "we welcome feedback"

_EGUsphere, 2023_

## Author Comment (AC1)

We thank the reviewer for his/her insightful comments. This is most appreciated and will help improve the revised version of the MS.

The key concern is about the constraints of the input parameters in our model. Specifically, concerns are raised about the inferred tectonic stress from the fracture orientation without properly describing the fractures. We acknowledge that providing a comprehensive description of the fractures would increase our understanding of the rationale behind our tectonic stress inference. We will include this aspect in the revised version. However, as part of this project, another paper dealt with the background fractures at the Latemar Carbonate Platform (N. Italy). There, we comprehensively described the fracture geometries, kinematics, driving factors, and connectivity. This paper is presently under revision.

| Specific comments/questions | Response |
|---|---|
| The presented investigation uses the outcropping network geometry as input for geomechanical and flow models, that is fair, but a key question could regard the timing of fractures formation.

In my opinion the authors must clarify and show the evidence that relate the fracture formation to time.
1. what is the evidence that exclude very late (i.e. during exhumation) formation of some of the observed fractures?

2. Since no crosscutting relationship are analysed, why should we think that all fractures experienced both tectonic stress regimes? | We used the outcropping network geometry as input for geomechanical and flow models.

We agree that capturing the timing of the fracture formation is essential. However, for this study, our model did not consider the formation and/or growth timing of new fractures in the study area. Instead, it considered that the already-developed fractures were either sheared, opened, and/or closed during the tectonic episodes.

In the Latemar, the two major tectonic events are associated with subsidence-related deformation in the Late Triassic and Early Jurassic times, shortly after the fractures were formed. This means that most of the fractures analysed were affected by this tectonic episode. In addition, a later Alpine compression during the Neogene overprinted the whole fracture network. This means new fractures may have formed during and/or after these tectonic episodes, including during the late exhumation.

However, the formation and growth of these new fractures were not mimicked in our model and are clearly beyond our model and study scope. We only focused on already-formed fractures, as they are today.

Detailed crosscutting relationships of the fractures were analysed in Igbokwe et al. (2022), and we will reflect some of this analysis in the revised version. |
| Line 255 "the arrangements orientations and the stress fields during the development of the fractures are documented". Stress inversion technique must be explained here. | We will explain the stress inversion technique. |
| Line 257 "In the Valsorda Valley (Fig. 3), carbonate outcrops are affected by minor reverse | |

| | |
|---|---|
| conjugate faults dipping at low angle (< 30o) to bedding." I strongly suggest the authors to clearly show displacements and kinematic indicators supporting this. | We agree and will show the displacement and kinematic indicators relating to the minor reverse conjugate faults. |
| Line 262 "On the other side, at the flat-topped Latemar, on the sub-horizontal (pavement) outcrops, fractures also form conjugate patterns, exhibiting dextral and sinistral displacements (Figs. 3 and 5)." I think that authors should highlight this evidence in figure 3 and 5.
Line 286 "Overall, two deformation phases were observed and documented" I think that authors must better document this. | Yes, that is a good point; we will highlight the evidence and document further the overall two deformation phases observed in the study area. |
| Line 298 "Although stylolite tends to hinder fluid flow (Boersma et al., 2019), observations in figures 3, 4 and 5 show they can enhance fluid movement". Even if I do think that stylolite generally hinder fluid flow I would suggest reading the paper by Heap et al., 2018 were it is proposed that stylolites can be considered as conduits for flow. I would also ask the authors to clearly describe and show the observations in figures 3, 4 and 5 proving that they can enhance fluid movement. | We agree with Heap et al. (2018) that stylolites can serve as fluid conduits.

Stylolites and fluid flow will be detailed in Figures 3 through 5.

However, for the geomechanical modelling (or numerical workflow), we treated all modelled features as reactivated fractures that control the fluid flow. In section 5.3 (Implication) of our work, we advanced the argument that "because of the effects of weathering and exhumation, the distinction between open fractures, veins, stylolites and shear fractures was not made in the model. Making these distinctions in our modelling workflow will complicate the whole process and is beyond the scope of this study. |
| Line 321 "Slight modifications and/or extrapolations of the fracture's original pattern were implemented to maintain the fracture topological connectivity" I would ask the authors to be more specific . | The revised section will provide more detail. Thank you for pointing this out. |
| Line 338 "When the fracture is in a closure condition, and sufficient loading is acting in the tangential direction, the fracture may slip. The slip and stick conditions of the fracture are determined based on the classical Coulomb's friction law". Coulomb friction depends on several parameters and it can greatly vary form host rock to host rock (see Collettini et al., 2019). It would be appropriate to have more information about the adopted mechanical properties. | Yes, that is a good point. That must be detailed in the revised version of this paper. |
| Line 411 "We have adopted the matrix permeability value of 2 x 10-15 m "Are there any | |

| | |
|---|---|
| other constrain eventually based of data from actual samples of the Latemar area or from similar lithologies? | We adopted the matrix permeability value of 2 x $10^{-15}$ m from the results of Whitaker et al. (2014) based on carbonate samples from Latemar.

Generally, carbonate rocks have very low matrix permeability with average values ranging from 2 to 4 x $10^{-15}$ $m^2$. We will detail the rationale based on other similar lithologies and explain why we chose the matrix permeability value we used |
| Line 437 "These tectonic episodes are constrained to the NW-SE and N-S shortening (compressive) directions with an assumed maximum magnitude of 50 and 160 MPa for the subsidence-related and Alpine deformation stages, respectively." I can't really see how authors constrain these stress values. Authors successively state that they refer to the The World Stress Map database, however as far as I know that paper is more focused on stress orientation that stress magnitude, moreover here there is no mention about the assumed depth so I found these stress values not clear. An explanation is given only at line 625, I think that it would be better to report and exlpain here the used input values. | We will move the explanation regarding the input parameters from line 625 to line 437.

The paper is focused more on the impact of the far-field stress orientation rather than stress magnitude values, which also affect the final effective permeability calculations.

However, the stress values mentioned here only indicate approximations from similar study areas with the same geodynamic conditions. This means that changing these values can change the final calculation of the effective permeability. Still, the workability of the model remains stable and robust and can compute any given matter.

Secondly, the numerical analysis (or modelling) is in the 2-D, constraining the effect of lateral expansion resulting from the overburden stress of the fracture domain. Considering depth implicates overburden stress. We advanced this argument in lines 828 – 839. |
| Line 445 "The constitutive parameter values are Youngs modulus 25 GPa and Poisson ration 0.30 (see Table 1 for detailed parameters)." Once again I would like the authors to specify where these input data come from. Moreover, if perfect elasticity is supposed in the model and dynamic parameters are used, they can differ a lot from static ones in particular for dolostone samples (see Trippetta et al., 2013). In any case a reference for such used values must be given. | We agree with the reviewers and will specify where the input parameters came from with relevant citations. |
| **Technical corections**

Scales are absent in figures 1d and e | Consider it done. |
| I have a problem also with the scales in figures 3 A and B. According to the shown scales outcrops are ~ 500 m long each. It is correct? | Needs to be verified. |

| | |
|---|---|
| Moreover the legend must be improved, I can't really distinguish between Tectonic styliolites and Perp. fractures

Line 255 and sub-horizontal (pavement) outcrops. Maybe "that" is missing here | Yes, thank you, we will do. |

---

## Author Comment (AC2)

We thank the reviewer for his insightful comments. This is most appreciated and will help improve the revised version of the MS.

The reviewer had raised two major and three minor concerns.

| Specific comments/questions | Response |
|---|---|
| **2 – Major items:**

Introduction (Lines 156-159): Previous simulation works taking into account superimposed tectonics events have "*not been well-represented*". What do the authors mean here? Do they mean that no study was carried out on the topic or that the quality of the previous studies is too poor? Please be more specific by introducing few references and by explaining why those studies were not properly carried out. | We will expand and explain in more detail the gap in the knowledge and/or in the literature with more literature cited. |
| Line: 328: Simulations are run in the 2-D domain. Please specify this in the abstract.

Line 341, Equation 3: Please specify the friction coefficient(s) assigned to the fractures and ideally, justify these values based on published works. | Consider it done. |
| Line 411: The matrix permeability is assumed constant and equal to 2 mD based on Whitaker et al. (2014)'s work. However, this later work shows that a large number of deposits have a permeability ranging between 50 mD and 5000mD (Fig. 7 in Whitaker et al). Could the authors explain selecting such low value?

Besides, the mean effective permeability of each FSS ranges between 0.85 and 3.24 mD (see Table 2 and Lines 608-613 in the submitted manuscript). Does it mean that fractures in the platform top have minor to negligible impact on overall bulk rock permeability if considering higher permeability for the host rock? Is there any published permeability measurement from the Latemar platform? If so, please add it in the methodology section for comparison purposes. | Thank you for pointing this out.

In the revised version, we will explain the rationale for selecting the low matrix permeability value in detail.

Carbonate rocks generally have very low matrix permeability with average values ranging from 2 to $4 \times 10^{-15}$ m$^2$.

We acknowledge that in some places, due to diagenetic, the matrix permeability of different carbonate lithologies can be very high, reaching up to 5 D. See Whitaker et al. (2014).

We constrained our values to 2 mD, which is only an indicative approximation, judging from similar study areas with the same geodynamic conditions.

This means that changing these values can change the final calculation of the effective permeability. Still, the workability of the model |

| | remains stable and robust and can compute any given matter. |
| --- | --- |
| | Assigning higher matrix permeability values at the platform top would increase the bulk permeability. For this reason, we used an average matrix permeability value to gauge the impact stress on fractures within the Latemar platform.

We will add other permeability measurements for the Latemar platform in the methodology section with relevant citations. |
| In the methodology section 3.1 (Line 295): The magnitude of the two far-field stress regimes is assumed. Which criteria these assumptions are based on? | See the above response for matrix permeability. To clarify further, the stress magnitude values mentioned here only indicate approximations from similar study areas with the same geodynamic conditions. This means that changing these values can change the final calculation of the effective permeability. Again, the workability of the model remains stable and robust and can compute any given matter.
In the revised version, we will explain this better as the rationale or criteria behind our assumption. |
| In line 475, it is mentioned that those boundary conditions are "feasible" for modelling compressive settings. Please be more specific on what "feasible" means here. | Thank you. We will be more specific in the revised version. |
| Stylolites are assumed as flow conduits despite controversy on the topic as stated in Line 297. The simulation results and conclusions on the enhanced bulk permeability caused by stress-induced fracture opening (Line 877) are thus optimistic and this should be stressed in the conclusion.
Note that the assumed low permeability of the host rock (2 mD) as discussed in the previous comments also adds to the overall key impact of fracture on increased permeability reported in the submitted manuscript. | Good point. Will be detailed in the revised version of this paper. |
| **3 – Minor items:**

Lines 122-124: Odd grammatical structure of the sentence. Please rephrase. | Consider it done. |
| Lines 125-126: If possible, please add any practical examples of change in the flow pattern | Good point. The revised version will provide more detail. |

| | |
|---|---|
| of reservoirs or storage sites caused by stress changes during injection/extraction of fluids. This would strengthen the relevancy of this work. | |
| Section 3.1.1: The methodology to collect data on fractures are well described.

What about the stylolites? Were they also collected using the same methods? Could you add information about this in this section? | Thank you.

Yes, we will add information about how stylolite data were collected.

However, for the geomechanical modelling (or numerical workflow), we treated all modelled features as reactivated fractures that control the fluid flow. We advanced the argument that "because of the effects of weathering and exhumation, the distinction between open fractures, veins, stylolites and shear fractures was not made in the model. Making these distinctions in our modelling workflow will complicate the whole process and is beyond the scope of this study. |
| Line 413: Add full stop at the end of the sentence. | Consider it done. |
| Fig 8: Do the minimum and maximum horizontal stress equal? I have not seen information about it so far. If both stresses are different, it would be good to add a plot displaying the evolution of $S_h$ and $S_H$ with time throughout the simulation period. | Thank you for pointing this out. The minimum and maximum horizontal stresses are not equal. We maintained the minimum horizontal stress value as zero and/or negligible value throughout the simulation period and gradually increased the maximum horizontal stress in load steps, representing Pseudo Time Steps (PST), analogues to quasi-static loading, until a maximum magnitude value is reached. See Figure 8c, showing the evolution of $S_H$ with time throughout the simulation period. |
| There are three publications of Bisdom et al., 2016. The reference "Bisdom et al., 2016c" is missing in the manuscript (see Lines 672, 741, 747, and 799). | Thanks, consider it done. |

---

## Author Response (AR1)

**Reply on RC1**

We thank the reviewer for his insightful comments. This is most appreciated and will help improve the revised version of the MS.

The reviewer had raised 2-major and 3-minor concerns, respectively, which we have addressed in the table below.

| Specific comments/questions | Response |
|---|---|
| **2 – Major items:**

Introduction (Lines 156-159): Previous simulation works taking into account superimposed tectonics events have *"not been well-represented"*. What do the authors mean here? Do they mean that no study was carried out on the topic or that the quality of the previous studies is too poor? Please be more specific by introducing few references and by explaining why those studies were not properly carried out. | Thanks, this is a good point; we have explained in more detail the gap in knowledge and/or in the literature and have included more references and citations (see lines 158 – 166) |
| Line: 328: Simulations are run in the 2-D domain. Please specify this in the abstract.

Line 341, Equation 3: Please specify the friction coefficient(s) assigned to the fractures and ideally, justify these values based on published works. | Done

Our model assumes that the frictional coefficient of the fracture plane is zero (see lines 384-385) We have also included this in Table 1 and also provided some references. |
| Line 411: The matrix permeability is assumed constant and equal to 2 mD based on Whitaker et al. (2014)'s work. However, this later work shows that many deposits have a permeability ranging between 50 mD and 5000mD (Fig. 7 in Whitaker et al). Could the authors explain selecting such low value?

Besides, the mean effective permeability of each FSS ranges between 0.85 and 3.24 mD (see Table 2 and Lines 608-613 in the submitted manuscript). Does it mean that fractures in the platform top have minor to negligible impact on overall bulk rock permeability if considering higher permeability for the host rock? Is there any published permeability measurement from the Latemar platform? If so, please add it in the methodology section for comparison purposes. | Thank you for pointing this out.

We have explained the rationale for selecting the low matrix permeability value (see lines 455-465)

Carbonate rocks generally have very low matrix permeability with average values ranging from 1 to 4 x $10^{-15}$ m$^2$.
We acknowledge that in some places, due to diverse (diagenetic) alterations (dissolution/ precipitation), the matrix permeability of different lithologies carbonate rocks can be very high, reaching up to 5 D, as noted in the simulated permeability values in some carbonate lithologies in Latemar addressed in Whitaker et al. (2014).
But, in this study, we constrained our values to 2 mD, which is only indicative approximations, judging from similar study areas with the same geodynamic conditions. |

| | This means that changing these values can change the final calculation of the effective permeability. Still, the workability of the model remains stable and robust and can compute any given matter. |
| | |
| | By assigning higher matrix permeability values at the platform top, we would expect an increase the bulk permeability. Nevertheless, the calculated permeability values here depend on fracture direction and interconnectivity. This means that the alignment of fractures can make the structure more permeable in a diagonal direction or vice versa, as can be noted from Table 2. We only used an average matrix permeability value of 2 mD to gauge the impact stress on fractures within the Latemar platform. |
| | |
| | Yes, we have added other published permeability values with respect to the Latemar platform in lines (460-461). |
| In the methodology section 3.1 (Line 295): The magnitude of the two far-field stress regimes is assumed. Which criteria these assumptions are based on? | We have explained the criteria for our assumption (see lines 484-505). Essentially, the stress magnitude values mentioned here are only indicate approximations from similar study areas with the same geodynamic conditions. This means that changing these values can change the final calculation of the effective permeability. Again, the workability of the model remains stable and robust and can compute any given matter. |
| In line 475, it is mentioned that those boundary conditions are "feasible" for modelling compressive settings. Please be more specific on what "feasible" means here. | We have reworded this sentence to highlight better our intentions (see lines 493-495). |
| Stylolites are assumed as flow conduits despite controversy on the topic as stated in Line 297. The simulation results and conclusions on the enhanced bulk permeability caused by stress-induced fracture opening (Line 877) are thus optimistic and this should be stressed in the conclusion. Note that the assumed low permeability of the host rock (2 mD) as discussed in the previous comments also adds to the overall key impact of fracture on increased permeability reported in the submitted manuscript. | Yes, that is a good point, and we have done that (see lines 948-953). |

| | |
|---|---|
| **3 – Minor items:**

Lines 122-124: Odd grammatical structure of the sentence. Please rephrase.

Lines 125-126: If possible, please add any practical examples of change in the flow pattern of reservoirs or storage sites caused by stress changes during injection/extraction of fluids. This would strengthen the relevancy of this work. | Done.

We have added a reference, pointing to the example of how changes in the flow pattern of reservoirs or storage sites are caused by stress changes during injection/extraction of fluids (see lines 123-126). |
| Section 3.1.1: The methodology to collect data on fractures are well described.

What about the stylolites? Were they also collected using the same methods? Could you add information about this in this section? | Thank you.

Yes, the information about collected stylolite data has been added (see section 3.1)

It is essential to note that for the geomechanical modelling (or numerical workflow), we treated all modelled features as reactivated fractures that control the fluid flow. We advanced the argument that "because of the effects of weathering and exhumation, the distinction between open fractures, veins, stylolites and shear fractures was not made in the model. Making these distinctions in our modelling workflow will complicate the whole process and is beyond the scope of this study. |
| Line 413: Add full stop at the end of the sentence. | Done. |
| Fig 8: Do the minimum and maximum horizontal stress equal? I have not seen information about it so far. If both stresses are different, it would be good to add a plot displaying the evolution of $S_h$ and $S_H$ with time throughout the simulation period. | Thank you for pointing this out. The minimum and maximum horizontal stresses are not equal. We maintained the minimum horizontal stress value as "zero" throughout the simulation period and gradually increased the maximum horizontal stress in load steps, representing Pseudo-Time Steps (PST), analogues to quasi-static loading, until a maximum magnitude value was reached.

We have now modified Figure 8c to show (i) the evolution of $S_H$ with time throughout the simulation period and (ii) the zero value of $S_h$ |

| | |
|---|---|
| | |
| There are three publications of Bisdom et al., 2016. The reference "Bisdom et al., 2016c" is missing in the manuscript (see Lines 672, 741, 747, and 799). | Thanks, we have included the references. |

**Reply on RC2**

We thank the reviewer for his/her insightful comments. This is most appreciated and will help improve the revised version of the MS.

The key concern is about the constraints of the input parameters in our model. Specifically, concerns are raised about the inferred tectonic stress from the fracture orientation without properly describing the fractures. We acknowledge that providing a comprehensive description of the fractures would increase our understanding of the rationale behind our tectonic stress inference. We have included some aspects of the fracture description in the revised version. However, as part of this project, another paper dealt with the background fractures at the Latemar Carbonate Platform (N. Italy). There, we comprehensively described the fracture geometries, kinematics, driving factors, and connectivity. This paper is presently under revision.

| Specific comments/questions | Response |
|---|---|
| The presented investigation uses the outcropping network geometry as input for geomechanical and flow models, that is fair, but a key question could regard the timing of fractures formation. | We used the outcropping network geometry as input for geomechanical and flow models. |
| In my opinion the authors must clarify and show the evidence that relate the fracture formation to time. | We agree that capturing the timing of the fracture formation is important. However, for this study, our model did not consider the formation and/or growth timing of new fractures in the study area. Instead, it considered that the already-developed fractures were either sheared, opened, and/or closed during the tectonic episodes (see lines 174-178) |
| 1. what is the evidence that exclude very late (i.e. during exhumation) formation of some of the observed fractures? | |
| 2. Since no crosscutting relationship are analysed, why should we think that all fractures experienced both tectonic stress regimes? | In the Latemar, the two major tectonic events are associated with subsidence-related deformation in the Late Triassic and Early Jurassic times, shortly after the fractures were formed. This means that most of the fractures analysed were affected by this tectonic episode. In addition, a later Alpine compression during the Neogene overprinted the whole fracture network. This means new fractures may have formed during and/or after these tectonic episodes, including during the late exhumation. |
| | However, the formation and growth of these new fractures were not mimicked in our model and are clearly beyond our model and study scope. We only focused on already-formed fractures, as they are today. |

| | |
|---|---|
| Line 255 "the arrangements orientations and the stress fields during the development of the fractures are documented". Stress inversion technique must be explained here. | We have explained the general assumption of stress state and stress inversion technique (see lines 263- 270) |

| | |
|---|---|
| Line 257 "In the Valsorda Valley (Fig. 3), carbonate outcrops are affected by minor reverse conjugate faults dipping at low angle (< 30o) to bedding." I strongly suggest the authors to clearly show displacements and kinematic indicators supporting this. | Done (see figure 3b and c) |
| Line 262 "On the other side, at the flat-topped Latemar, on the sub-horizontal (pavement) outcrops, fractures also form conjugate patterns, exhibiting dextral and sinistral displacements (Figs. 3 and 5)." I think that authors should highlight this evidence in figure 3 and 5.
Line 286 "Overall, two deformation phases were observed and documented" I think that authors must better document this. | Yes, that is a good point; we have highlighted the kinematic indicator and documented better the overall deformation phases observed in the study area (see Figure 4, and Section 3.1) |
| Line 298 "Although stylolite tends to hinder fluid flow (Boersma et al., 2019), observations in figures 3, 4 and 5 show they can enhance fluid movement". Even if I do think that stylolite generally hinder fluid flow I would suggest reading the paper by Heap et al., 2018 were it is proposed that stylolites can be considered as conduits for flow. I would also ask the authors to clearly describe and show the observations in figures 3, 4 and 5 proving that they can enhance fluid movement. | We agree with Heap et al. (2018) that stylolites can serve as fluid conduits.

In Figures 3 and 4, we have documented reactivated stylolites, which acted as fluid flow conduits.

However, for the geomechanical modelling (or numerical workflow), we treated all modelled features as (reactivated) fractures that control the fluid flow. In section 5.3 (Implication) of our work, we advanced the argument that "because of the effects of weathering and exhumation, the distinction between open fractures, veins, stylolites and shear fractures was not made in the model. Making these distinctions in our modelling workflow will complicate the whole process and is beyond the scope of this study. |
| Line 321 "Slight modifications and/or extrapolations of the fracture's original pattern were implemented to maintain the fracture topological connectivity" I would ask the authors to be more specific . | We have struck out this sentence in the revised version. Our model did not modify the original fracture pattern. Thank you for pointing this out. |

| | |
|---|---|
| Line 338 "When the fracture is in a closure condition, and sufficient loading is acting in the tangential direction, the fracture may slip. The slip and stick conditions of the fracture are determined based on the classical Coulomb's friction law". Coulomb friction depends on several parameters and it can greatly vary form host rock to host rock (see Collettini et al., 2019). It would be appropriate to have more information about the adopted mechanical properties. | Yes, that is a good point. We have added more information on the adopted mechanical properties (see lines 385-386 and Table 1 for details). |

| | |
|---|---|
| Line 411 "We have adopted the matrix permeability value of 2 x 10-15 m "Are there any other constrain eventually based of data from actual samples of the Latemar area or from similar lithologies? | We adopted the matrix permeability value of 2 x $10^{-15}$ m from the results of Whitaker et al. (2014) based on carbonate samples from Latemar.

Generally, carbonate rocks have very low matrix permeability with average values ranging from 1 to 4 x $10^{-15}$ $m^2$. We have detailed the rationale for adopting 2 mD as the matrix permeability value (see lines 457 – 465) |
| Line 437 "These tectonic episodes are constrained to the NW-SE and N-S shortening (compressive) directions with an assumed maximum magnitude of 50 and 160 MPa for the subsidence-related and Alpine deformation stages, respectively." I can't really see how authors constrain these stress values. Authors successively state that they refer to the The World Stress Map database, however as far as I know that paper is more focused on stress orientation that stress magnitude, moreover here there is no mention about the assumed depth so I found these stress values not clear. An explanation is given only at line 625, I think that it would be better to report and exlpain here the used input values. | We have moved the explanation regarding the input parameters from line 625 to line 485-505.

The paper is focused more on the impact of the far-field stress orientation rather than stress magnitude values, which also affect the final effective permeability calculations.

However, the stress values mentioned here are only indicate approximations (see lines 485-505). It is important to note that changing these values can change the final calculation of the effective permeability. Still, the workability of the model remains stable and robust and can compute any given matter.

Secondly, the numerical analysis (or modelling) is in the 2-D, thus, constraining the effect of lateral expansion resulting from the overburden stress of the fracture domain. Considering depth implicates overburden stress. We advanced this argument in lines 893 – 899. |

| | |
|---|---|
| Line 445 "The constitutive parameter values are Youngs modulus 25 GPa and Poisson ration 0.30 (see Table 1 for detailed parameters)." Once again I would like the authors to specify where these input data come from. Moreover, if perfect elasticity is supposed in the model and dynamic parameters are used, they can differ a lot from static ones in particular for dolostone samples (see Trippetta et al., 2013). In any case a reference for such used values must be given. | We have specified and cited the reference showing where the input parameters were adopted (see Table 1, and lines 384-385). |
| **Technical corections**

Scales are absent in figures 1d and e

I have a problem also with the scales in figures 3 A and B. According to the shown scales outcrops are ~ 500 m long each. It is correct? | We have included the explanations for the scale in the figure title. The red arrows indicate the length and height of the scale.

We have verified and modified the scale accordingly. |
| Moreover the legend must be improved, I can't really distinguish between Tectonic styliolites and Perp. fractures

Line 255 and sub-horizontal (pavement) outcrops. Maybe "that" is missing here | Done. |